# SYMMETRIC WASSERSTEIN AUTOENCODERS

## ABSTRACT

Leveraging the framework of Optimal Transport, we introduce a new family of generative autoencoders with a learnable prior, called Symmetric Wasserstein Autoencoders (SWAEs). We propose to symmetrically match the joint distributions of the observed data and the latent representation induced by the encoder and the decoder. The resulting algorithm jointly optimizes the modelling losses in both the data and the latent spaces with the loss in the data space leading to the denoising effect. With the symmetric treatment of the data and the latent representation, the algorithm implicitly preserves the local structure of the data in the latent space. To further improve the latent representation, we incorporate a reconstruction loss into the objective, which significantly benefits both the generation and reconstruction. We empirically show the superior performance of SWAEs over the state-of-the-art generative autoencoders in terms of classification, reconstruction, and generation.

## 1 INTRODUCTION

Deep generative models have emerged as powerful frameworks for modelling complex data. Widely used families of such models include Generative Adversarial Networks (GANs) (Goodfellow et al., 2014), Variational Autoencoders (VAEs) (Rezende et al., 2014; Kingma & Welling, 2014), and autoregressive models (Uria et al., 2013; Van Oord et al., 2016). The VAE-based framework has been popular as it yields a bidirectional mapping, *i.e.*, it consists of both an inference model (from data to latent space) and a generative model (from latent to data space). With an inference mechanism VAEs can provide a useful latent representation that captures salient information about the observed data. Such latent representation can in turn benefit downstream tasks such as clustering, classification, and data generation. In particular, the VAE-based approaches have achieved impressive performance results on challenging real-world applications, including image synthesizing (Razavi et al., 2019), natural text generation (Hu et al., 2017), and neural machine translation (Sutskever et al., 2014).

VAEs aim to maximize a tractable variational lower bound on the log-likelihood of the observed data, commonly called the ELBO. Since VAEs focus on modelling the marginal likelihood of the data instead of the joint likelihood of the data and the latent representation, the quality of the latent is not well assessed (Alemi et al., 2017; Zhao et al., 2019), which is undesirable for learning useful representation. Besides the perspective of maximum-likelihood learning of the data, the objective of VAEs is equivalent to minimizing the KL divergence between the encoding and the decoding distributions, with the former modelling the joint distribution of the observed data and the latent representation induced by the encoder and the latter modelling the corresponding joint distribution induced by the decoder. Such connection has been revealed in several recent works (Livne et al., 2019; Esmaeili et al., 2019; Pu et al., 2017b; Chen et al., 2018). Due to the asymmetry of the KL divergence, it is highly likely that the generated samples are of a low probability in the data distribution, which often leads to unrealistic generated samples (Li et al., 2017b; Alemi et al., 2017).

A lot of work has proposed to improve VAEs from different perspectives. For example, to enhance the latent expressive power VampPrior (Tomczak & Welling, 2018), normalizing flow (Rezende & Mohamed, 2015), and Stein VAEs (Pu et al., 2017a) replace the Gaussian distribution imposed on the latent variables with a more sophisticated and flexible distribution. However, these methods are all based on the objective of VAEs, which therefore are unable to alleviate the limitation of VAEs induced by the objective. To improve the latent representation (Zhao et al., 2019) explicitly includes the mutual information between the data and the latent into the objective. Moreover, to address the asymmetry of the KL divergence in VAEs (Livne et al., 2019; Chen et al., 2018; Pu et al., 2017b) leverage a symmetric divergence measure between the encoding and the decoding distributions.

Nevertheless, these methods typically involve a sophisticated objective function that either depends on unstable adversarial training or challenging approximation of the mutual information.

In this paper, we leverage Optimal Transport (OT) (Villani, 2008; Peyré et al., 2019) to symmetrically match the encoding and the decoding distributions. The OT optimization is generally challenging particularly in high dimension, and we address this difficulty by transforming the OT cost into a simpler form amenable to efficient numerical implementation. Owing to the symmetric treatment of the observed data and the latent representation, the local structure of the data can be implicitly preserved in the latent space. However, we found that with the symmetric treatment only the performance of the generative model may not be satisfactory. To improve the generative model we additionally include a reconstruction loss into the objective, which is shown to significantly benefit the quality of the generation and reconstruction.

Our contributions can be summarized as follows. Firstly, we propose a new family of generative autoencoders, called Symmetric Wasserstein Autoencoders (SWAEs). Secondly, we adopt a learnable latent prior, parameterized as a mixture of the conditional priors given the learnable pseudo-inputs, which prevents SWAEs from over-regularizing the latent variables. Thirdly, we empirically perform an ablation study of SWAEs in terms of the KNN classification, denoising, reconstruction, and sample generation. Finally, we empirically verify, using benchmark tasks, the superior performance of SWAEs over several state-of-the-art generative autoencoders.

## 2 SYMMETRIC WASSERSTEIN AUTOENCODERS

In this section we introduce a new family of generative autoencoders, called Symmetric Wasserstein Autoencoders (SWAEs).

### 2.1 OT FORMULATION

Denote the random vector at the encoder as $\mathbf{e} \triangleq (\mathbf{x}_e, \mathbf{z}_e) \in \mathcal{X} \times \mathcal{Z}$, which contains both the observed data $\mathbf{x}_e \in \mathcal{X}$ and the latent representation $\mathbf{z}_e \in \mathcal{Z}$. We call the distribution $p(\mathbf{e}) \sim p(\mathbf{x}_e)p(\mathbf{z}_e|\mathbf{x}_e)$ *the encoding distribution*, where $p(\mathbf{x}_e)$ represents the data distribution and $p(\mathbf{z}_e|\mathbf{x}_e)$ characterizes an inference model. Similarly, denote the random vector at the decoder as $\mathbf{d} \triangleq (\mathbf{x}_d, \mathbf{z}_d) \in \mathcal{X} \times \mathcal{Z}$, which consists of both the latent prior $\mathbf{z}_d \in \mathcal{Z}$ and the generated data $\mathbf{x}_d \in \mathcal{X}$. We call the distribution $p(\mathbf{d}) \sim p(\mathbf{z}_d)p(\mathbf{x}_d|\mathbf{z}_d)$ *the decoding distribution*, where $p(\mathbf{z}_d)$ represents the prior distribution and $p(\mathbf{x}_d|\mathbf{z}_d)$ characterizes a generative model. The objective of VAEs is equivalent to minimizing the (asymmetric) KL divergence between the encoding distribution $p(\mathbf{e})$ and the decoding distribution $p(\mathbf{d})$ (see Appendix A.1). To address the limitation in VAEs, first we propose to treat the data and the latent representation symmetrically instead of asymmetrically by minimizing the $p$-th Wasserstein distance between $p(\mathbf{e})$ and $p(\mathbf{d})$ leveraging Optimal Transport (OT) (Villani, 2008; Peyré et al., 2019).

OT provides a framework for comparing two distributions in a Lagrangian framework, which seeks the minimum cost for transporting one distribution to another. We focus on the primal problem of OT, and Kantorovich's formulation (Peyré et al., 2019) is given by:

$$W_c(p(\mathbf{e}), p(\mathbf{d})) \triangleq \inf_{\Gamma \in \mathcal{P}(\mathbf{e} \sim p(\mathbf{e}), \mathbf{d} \sim p(\mathbf{d}))} \mathbb{E}_{(\mathbf{e}, \mathbf{d}) \sim \Gamma} \quad c(\mathbf{e}, \mathbf{d}), \tag{1}$$

where $\mathcal{P}(\mathbf{e} \sim p(\mathbf{e}), \mathbf{d} \sim p(\mathbf{d}))$, called the coupling between $\mathbf{e}$ and $\mathbf{d}$, denotes the set of the joint distributions of $\mathbf{e}$ and $\mathbf{d}$ with the marginals $p(\mathbf{e})$ and $p(\mathbf{d})$, respectively, and $c(\mathbf{e}, \mathbf{d}) : (\mathcal{X}, \mathcal{Z}) \times (\mathcal{X}, \mathcal{Z}) \to [0, +\infty]$ denotes the cost function. When $((\mathcal{X}, \mathcal{Z}) \times (\mathcal{X}, \mathcal{Z}), d)$ is a metric space and the cost function $c(\mathbf{e}, \mathbf{d}) = d^p(\mathbf{e}, \mathbf{d})$ for $p \geq 1$, $W_p$, the $p$-th root of $W_c$ is defined as the $p$-th Wasserstein distance. In particular, it can be proved that the $p$-th Wasserstein distance is a metric hence *symmetric*, and metrizes the weak convergence (see, *e.g.*, (Santambrogio, 2015)).

Optimization of equation 1 is computationally prohibitive especially in high dimension (Peyré et al., 2019). To provide an efficient solution, we restrict to the deterministic encoder and decoder. Specifically, at the encoder we have the latent representation $\mathbf{z}_e = E(\mathbf{x}_e)$ with the function $E : \mathcal{X} \to \mathcal{Z}$, and at the decoder we have the generated data $\mathbf{x}_d = D(\mathbf{z}_d)$ with the function $D : \mathcal{Z} \to \mathcal{X}$. It turns out that with the deterministic condition instead of searching for an optimal coupling in high dimension, we can find a proper conditional distribution $p(\mathbf{z}_d|\mathbf{x}_e)$ with the marginal $p(\mathbf{z}_d)$.

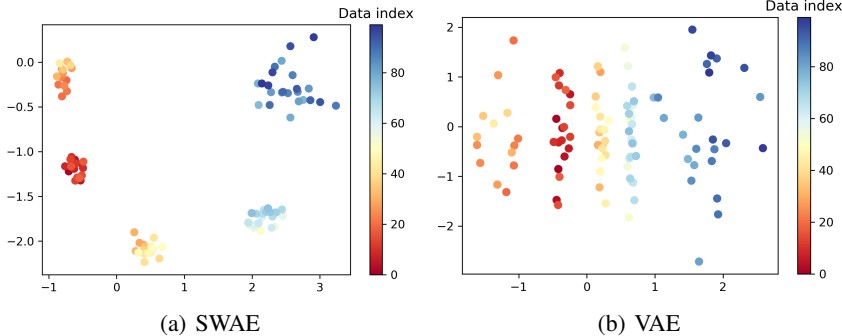

|   (a) SWAE   |   (b) VAE   |

Figure 1: Latent representations of 100 GMM samples (mode 5 and dimension 10) with dim-$\mathbf{z} = 2$. The indexes of these latent representations are sorted based on the distance to a target sample in the data space, *i.e.*, Index 0 is associated with the target sample and Index 100 is associated with the furthest sample to the target in the data space. With SWAE (left) data samples that are close in the data space are also close in the latent space, while VAE (right) cannot preserve such correspondence.

**Theorem 1** *Given the deterministic encoder $E : \mathcal{X} \to \mathcal{Z}$ and the deterministic decoder $D : \mathcal{Z} \to \mathcal{X}$, the OT problem in equation 1 can be transformed to the following:*

$$W_c(p(\mathbf{e}), p(\mathbf{d})) = \inf_{p(\mathbf{z}_d|\mathbf{x}_e)} \mathbb{E}_{p(\mathbf{x}_e)} \mathbb{E}_{p(\mathbf{z}_d|\mathbf{x}_e)} \quad c(\mathbf{e}, \mathbf{d}), \tag{2}$$

*where the observed data follows the distribution $p(\mathbf{x}_e)$ and the prior follows the distribution $p(\mathbf{z}_d)$.*

The proof of Theorem 1 extends that of Theorem 1 in (Tolstikhin et al., 2018), and is provided in Appendix A.2. If $\mathcal{X} \times \mathcal{Z}$ is the Euclidean space endowed with the $L_p$ norm, then the expression in equation 2 equals the following:

$$W_c(p(\mathbf{e}), p(\mathbf{d})) = \inf_{p(\mathbf{z}_d|\mathbf{x}_e)} \mathbb{E}_{p(\mathbf{x}_e)} \mathbb{E}_{p(\mathbf{z}_d|\mathbf{x}_e)} \quad \|\mathbf{x}_e - D(\mathbf{z}_d)\|_p^p + \|E(\mathbf{x}_e) - \mathbf{z}_d\|_p^p, \tag{3}$$

where in the objective we call the first term the **x**-loss and the second term the **z**-loss. With the above transformation, we decompose the loss in the joint space into the losses in both the data and the latent spaces. Such decomposition is crucial and allows us to treat the data and the latent representation symmetrically.

The **x**-loss, *i.e.*, $\|\mathbf{x}_e - D(\mathbf{z}_d)\|_p^p$, represents the discrepancy in the data space, and can be interpreted from two different perspectives. Firstly, since $D(\mathbf{z}_d)$ represents the generated data, the **x**-loss essentially minimizes the dissimilarity between the observed data and the generated data. Secondly, the **x**-loss is closely related to the objective of Denoising Autoencoders (DAs) (Vincent et al., 2008; 2010). In particular, DAs aim to minimize the discrepancy between the observed data and a partially destroyed version of the observed data. The corrupted data can be obtained by means of a stochastic mapping from the original data (*e.g.*, via adding noises). By contrast, the **x**-loss can be explained in the same way with the generated data being interpreted as the corrupted data. This is because the prior $\mathbf{z}_d$ in $D(\mathbf{z}_d)$ is sampled from the conditional distribution $p(\mathbf{z}_d|\mathbf{x}_e)$, which depends on the observed data $\mathbf{x}_e$. Consequently, the generated data $D(\mathbf{z}_d)$, obtained by feeding $\mathbf{z}_d$ to the decoder, is stochastically related to the observed data $\mathbf{x}_e$. With this insight, the same as the objective of DAs, the **x**-loss can lead to the denoising effect.

The **z**-loss, *i.e.*, $\|E(\mathbf{x}_e) - \mathbf{z}_d\|_p^p$, represents the discrepancy in the latent space. The whole objective in equation 3 hence simultaneously minimizes the discrepancy in the data and the latent spaces. Observe that in equation 3 $E(\mathbf{x}_e)$ is the latent representation of $\mathbf{x}_e$ at the encoder, while $\mathbf{z}_d$ can be thought of as the latent representation of $D(\mathbf{z}_d)$ at the decoder. With such connection, the optimization of equation 3 can preserve *the local data structure* in the latent space. More specifically, since $\mathbf{x}_e$ and $D(\mathbf{z}_d)$ are stochastically dependent, roughly speaking, if two data samples are close to each other in the data space, their corresponding latent representations are also expected to be close. This is due to the symmetric treatment of the data and the latent representation. In Figure 1 we illustrate this effect and compare SWAE with VAE.

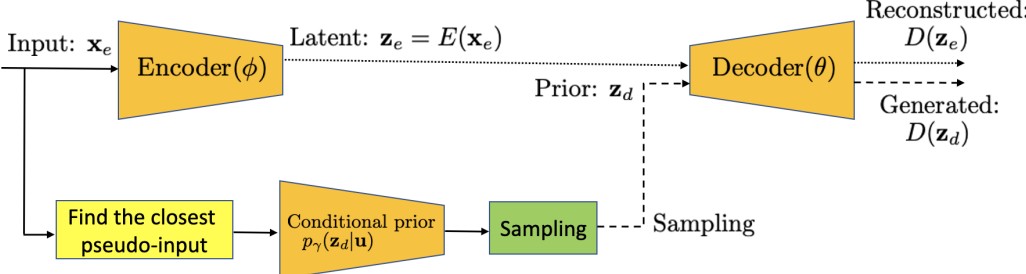

Figure 2: Network architecture of SWAEs. To generate new data latent samples are first drawn from the marginal prior $p(\mathbf{z}_d)$ based on the conditional priors $p(\mathbf{z}_d|\mathbf{u}_k)$, and then fed to the decoder.

**Comparison with WAEs** (Tolstikhin et al., 2018) The objective in equation 3 minimizes the OT cost between the joint distributions of the data and the latent, *i.e.*, $W_c(p(\mathbf{e}), p(\mathbf{d}))$, while the objective of WAEs (Tolstikhin et al., 2018) minimizes the OT cost between the marginal distributions of the data, *i.e.*, $W_c(p(\mathbf{x}_e), p(\mathbf{x}_d))$, where $p(\mathbf{x}_d)$ is the marginal data distribution induced by the decoding distribution $p(\mathbf{d})$. The problem of WAEs is first formulated as an optimization with the constraint $p(\mathbf{z}_e) = p(\mathbf{z}_d)$, where $p(\mathbf{z}_e)$ is the marginal distribution induced by the encoding distribution $p(\mathbf{e})$, and then relaxed by adding a regularizer. With the deterministic decoder, the final optimization problem of WAEs is as follows:

$$\inf_{p(\mathbf{z}_e|\mathbf{x}_e)} \mathbb{E}_{p(\mathbf{x}_e)} \mathbb{E}_{p(\mathbf{z}_e|\mathbf{x}_e)} \quad c(\mathbf{x}_e, D(\mathbf{z}_e)) + \lambda \mathcal{D}(p(\mathbf{z}_e), p(\mathbf{z}_d)), \tag{4}$$

where $\mathcal{D}(,)$ denotes some divergence measure. Comparing equation 4 to equation 3, we can see that both methods decompose the loss into the losses in the data and the latent spaces. Differently, in equation 4 the first term reflects the reconstruction loss in the data space and the second term represents the distribution-based dissimilarity in the latent space; while in equation 3 the $\mathbf{x}$-loss is closely related to the denoising and the generation quality and the $\mathbf{z}$-loss measures the sample-based dissimilarity. Moreover, equation 4 is optimized over the posterior $p(\mathbf{z}_e|\mathbf{x}_e)$ with a fixed prior $p(\mathbf{z}_d)$, while equation 3 is optimized over the conditional prior $p(\mathbf{z}_d|\mathbf{x}_e)$ with a potentially learnable prior.

## 2.2 IMPROVEMENT OF LATENT REPRESENTATION

The objective in equation 3 only seeks to match the encoding and the decoding distributions. Besides the encoder and the decoder structures, there is no explicit constraint on the correlation between the data and the latent representation within each joint distribution. Lacking of such constraint typically results in a low quality of reconstruction (Dumoulin et al., 2017; Li et al., 2017a). Therefore, we incorporate a reconstruction-based loss into the objective associated with a controllable coefficient. Additionally, since the dimension of the latent space is usually much smaller than that of the data space, we associate a weighting parameter to balance these two types of losses. Overall, the objective function can be represented as follows:

$$\inf_{p(\mathbf{z}_d|\mathbf{x}_e)} \mathbb{E}_{p(\mathbf{x}_e)} \mathbb{E}_{p(\mathbf{z}_d|\mathbf{x}_e)} \quad \beta\|\mathbf{x}_e - D(\mathbf{z}_d)\|_p^p + (1-\beta)\|\mathbf{x}_e - D(\mathbf{z}_e)\|_p^p + \alpha\|E(\mathbf{x}_e) - \mathbf{z}_d\|_p^p, \tag{5}$$

where $\|\mathbf{x}_e - D(\mathbf{z}_e)\|_p^p$ denotes the reconstruction loss, and $\beta(0 < \beta < 1)$ and $\alpha(\alpha > 0)$ are the weighting parameters. The weighting parameter $\beta$ controls the trade-off between the $\mathbf{x}$-loss and the reconstruction loss, and a smaller value of $\beta$ generally leads to better reconstruction. To achieve a better trade-off between the generation and reconstruction $\beta$ needs to be carefully chosen. We will perform an ablation study of SWAEs and show the importance of including the reconstruction loss into the objective for the generative model in Section 3.

## 2.3 ALGORITHM

Similar to many VAE-based generative models, we assume that the encoder, the decoder, and the conditional prior are parameterized by deep neural networks. Unlike the canonical VAEs, where the prior distribution is simple and given in advance, the proposed method adopts a learnable prior. The

benefits of a learnable prior, *e.g.*, avoiding over-regularization and hence improving the quality of the latent representation, have been revealed in several recent works (Hoffman & Johnson, 2016; Tomczak & Welling, 2018; Atanov et al., 2019; Klushyn et al., 2019). Obviously, the conditional prior is related to the marginal prior via $\mathbb{E}_{\mathbf{x}_e} p(\mathbf{z}_d|\mathbf{x}_e) = p(\mathbf{z}_d)$. This indicates a way to design the prior as a mixture of the conditional distributions, *i.e.*, $p^*(\mathbf{z}_d) = \frac{1}{N}\sum_{n=1}^{N} p(\mathbf{z}_d|\mathbf{x}_{e,n})$, where $\mathbf{x}_{e,1}, \cdots, \mathbf{x}_{e,N}$ are the training samples. To avoid over-fitting, similar to (Tomczak & Welling, 2018), we replace the training samples with learnable pseudo-inputs and parameterize the prior distribution $p(\mathbf{z}_d)$ as $p_\gamma(\mathbf{z}_d) = \frac{1}{K}\sum_{k=1}^{K} p_\gamma(\mathbf{z}_d|\mathbf{u}_k)$, where $\gamma$ denotes the parameters of the conditional prior network, $\mathbf{u}_k \in \mathcal{X}$ are the learnable pseudo-inputs, and $K$ is the number of the pseudo-inputs. We emphasize that the conditional prior $p(\mathbf{z}_d|\mathbf{x}_e)$ (or approximated $p(\mathbf{z}_d|\mathbf{u}_k)$) is used to obtain the marginal prior $p(\mathbf{z}_d)$; while the posterior $p(\mathbf{z}_e|\mathbf{x}_e)$ is used for inference. In experiment, we parameterize the conditional prior as a Gaussian distribution.

We call the proposed generative model Symmetric Wasserstein Autoencoders (SWAEs) as we treat the observed data and the latent representation symmetrically. We summarize the training algorithm in Algorithm 1 and show the network architecture in Figure 2. As an example, we define the cost function $c(,)$ as the squared $L2$ norm.

---

**Algorithm 1:** Symmetric Wasserstein Autoencoders (SWAEs)

---

**Require:** The number of the pseudo-inputs $K$. The weighting parameters $\beta$ and $\alpha$. Initialize the parameters $\phi, \theta$, and $\gamma$ of the encoder network, the decoder network, and the conditional prior network, respectively.

**while** $(\phi, \theta, \gamma, \{\mathbf{u}_k\})$ *not converged* **do**

    1. Sample $\{\mathbf{x}_{e,1}, \cdots, \mathbf{x}_{e,N}\}$ from the training dataset.

    2. Find the closest pseudo-input $\mathbf{u}^{(n)}$ of each training sample from the set $\{\mathbf{u}_1, \cdots, \mathbf{u}_K\}$.

    3. Sample $\mathbf{z}_{d,n}$ from the conditional prior $p_\gamma(\mathbf{z}_d|\mathbf{u}^{(n)})$ for $n = 1, \cdots, N$.

    4. Update $(\phi, \theta, \gamma, \{\mathbf{u}_k\})$ by descending the cost function

       $\frac{1}{N}\sum_{n=1}^{N} \beta\|\mathbf{x}_{e,n} - D(\mathbf{z}_{d,n})\|_2^2 + (1-\beta)\|\mathbf{x}_{e,n} - D(E(\mathbf{x}_{e,n}))\|_2^2 + \alpha\|E(\mathbf{x}_{e,n}) - \mathbf{z}_{d,n}\|_2^2.$

---

Since we use the pseudo-inputs instead of the training samples in the conditional prior, given each training sample we need to find the closest pseudo-input in Step 2. To measure the similarity, we can use, *e.g.*, the $L2$ norm or the cosine similarity. Since the dimension of the latent space is usually much smaller than that of the data space, to reduce the searching time we can alternatively perform Step 2 in the latent space as an approximation. Specifically, we can find the closest latent representation of $E(\mathbf{x}_{e,n})$ from the set $\{E(\mathbf{u}_1), \cdots, E(\mathbf{u}_K)\}$ so as to obtain the corresponding closest pseudo-input. From the experiment we found that such approximation results in little performance degradation, and we attribute it to the preservation of the local structure as explained before.

## 3 EXPERIMENTAL RESULTS

In this section, we compare the performance of the proposed SWAE with several contemporary generative autoencoders, namely VAE (Kingma & Welling, 2014), WAE-GAN (Tolstikhin et al., 2018), WAE-MMD (Tolstikhin et al., 2018), VampPrior (Tomczak & Welling, 2018), and MIM (Livne et al., 2019), using four benchmark datasets: MNIST, Fashion-MNIST, Coil20, and CIFAR10 with a subset of classes (denoted as CIFAR10-sub).

### 3.1 EXPERIMENTAL SETUP

The design of neural network architectures is orthogonal to that of the algorithm objective, and can greatly affect the algorithm performance (Vahdat & Kautz, 2020). Since MIM has the same network architecture as that of VampPrior, for fair comparison we also build SWAE as well as VAE based on the VampPrior network architecture. In particular, VampPrior adopts the hierarchical latent structure with the convolutional layers (*i.e.*, convHVAE ($L = 2$)), where the gating mechanism is utilized as an element-wise non-linearity. The building block of the network structure of VAE and SWAE is the same as that of VampPior except that the latent structure is non-hierarchical. Different from SWAE, the prior of VampPrior and MIM is designed as a mixture of the posteriors (instead

Table 1: Classification accuracy of 5-NN (averaged over 5 trials). The standard deviation is generally less than 0.01 and is omitted in the table.

| Dataset | dim-$\mathbf{z}$ | SWAE ($\beta = 1$) | SWAE ($\beta = 0.5$) | SWAE ($\beta = 0$) | VAE | WAE-GAN | WAE-MMD | VampPrior | MIM |
|---|---|---|---|---|---|---|---|---|---|
| | 8 | 0.96 | **0.97** | **0.97** | 0.96 | 0.87 | **0.97** | **0.97** | **0.97** |
| MNIST | 40 | **0.97** | **0.97** | **0.97** | 0.80 | 0.68 | 0.96 | 0.93 | **0.97** |
| | 80 | **0.97** | **0.97** | **0.97** | 0.60 | 0.90 | 0.94 | 0.86 | 0.96 |
| Fashion-MNIST | 8 | 0.82 | 0.81 | **0.83** | 0.80 | 0.71 | 0.80 | 0.81 | 0.82 |
| | 40 | **0.84** | 0.83 | **0.84** | 0.54 | 0.62 | 0.82 | 0.81 | 0.82 |
| | 80 | **0.84** | 0.83 | 0.83 | 0.37 | 0.54 | 0.76 | 0.74 | 0.81 |
| Coil20 | 8 | 0.95 | 0.97 | 0.97 | 0.95 | **0.98** | 0.78 | 0.91 | 0.89 |
| | 40 | 0.97 | 0.98 | 0.97 | 0.90 | **0.99** | **0.99** | 0.96 | 0.96 |
| | 80 | 0.97 | **0.98** | **0.98** | 0.97 | **0.98** | 0.98 | 0.96 | **0.98** |
| CIFAR10-sub | 80 | **0.69** | 0.67 | 0.65 | 0.67 | 0.61 | 0.68 | 0.68 | 0.66 |
| | 256 | **0.70** | 0.66 | 0.61 | 0.62 | 0.61 | 0.68 | 0.65 | 0.65 |
| | 512 | **0.70** | 0.66 | 0.62 | 0.55 | 0.60 | 0.68 | 0.64 | 0.66 |

of a mixture of the conditional priors as in SWAE) conditioned on the learnable pseudo-inputs. The pseudo-inputs in SWAE, VampPrior, and MIM are initialized with the training samples. For VampPrior and MIM, the number of the pseudo-inputs $K$ is carefully chosen via the validation set. Unlike these two algorithms, for SWAE we found that increasing $K$ improves the algorithm performance. The setup of $K$ for SWAE, VampPrior, and MIM on all datasets can be found in Appendix A.3. For SWAE, we set the weighting parameter $\alpha$ to 1 in all cases; in Step 2 we use the $L2$ norm as the similarity measure in the data space. WAE-GAN and WAE-MMD are the WAE-based models, where the divergence measure in the latent space is based on GAN and the maximum mean discrepancy (MMD), respectively. The network structure of WAE-GAN and WAE-MMD is the same as that used in (Tolstikhin et al., 2018). The prior of VAE, WAE-GAN, and WAE-MMD is set as an isotropic Gaussian. A detailed description of the datasets, the applied network architectures, and the training parameters can be found in Appendix A.3.

## 3.2 LATENT REPRESENTATION

The latent representation is expected to capture salient features of the observed data and be useful for the downstream applications. The considered datasets are all associated with the labels. In the experiment we use the latent representation for the K-Nearest Neighbor (KNN) classification and compare the classification accuracy of 5-NN in Table 1, where dim-$\mathbf{z}$ denotes the dimension of the latent space. The results of 3-NN and 10-NN are similar to those of 5-NN and thus are omitted. We found that the classification results of all algorithms on CIFAR10 are unsatisfactory based on the current networks (accuracy was around $0.3 - 0.4$; this may due to the limited expressive power of the shallow network architectures used), so instead we create a subset of CIFAR10 (CIFAR10-sub) which contains 3 classes: bird, cat, and ship.

Since the prior of VAE, WAE-GAN, and WAE-MMD is an isotropic Gaussian, setting dim-$\mathbf{z}$ greater than the intrinsic dimensionality of the observed data would force $p(\mathbf{z}_e)$ to be in a manifold in the latent space (Tolstikhin et al., 2018). This makes it impossible to match the marginal $p(\mathbf{z}_e)$ with the prior $p(\mathbf{z}_d)$ and thus leads to unsatisfactory latent representation. Such concern can be verified particularly on Fashion-MNIST where the classification accuracy of VAE and WAE-GAN drops dramatically when dim-$\mathbf{z}$ is increased. For SWAE, we consider two cases: $\beta = 1$ (*i.e.*, without the reconstruction loss) and $\beta = 0.5$. The classification accuracy of SWAE ($\beta = 1$) is comparable to SWAE ($\beta = 0.5$) and is generally superior for different values of dim-$\mathbf{z}$ to the benchmarks.

To further show the structure of the latent representation, we project the latent representation to 2D using t-SNE (Maaten & Hinton, 2008) as the visualization tool. As an example, we show the projection of the latent representation on MNIST in Figure 3. We can see that SWAEs keep the local structure of the observed data in the latent space and lead to tight clusters, which is consistent to our expectation as explained in Section 2.1.

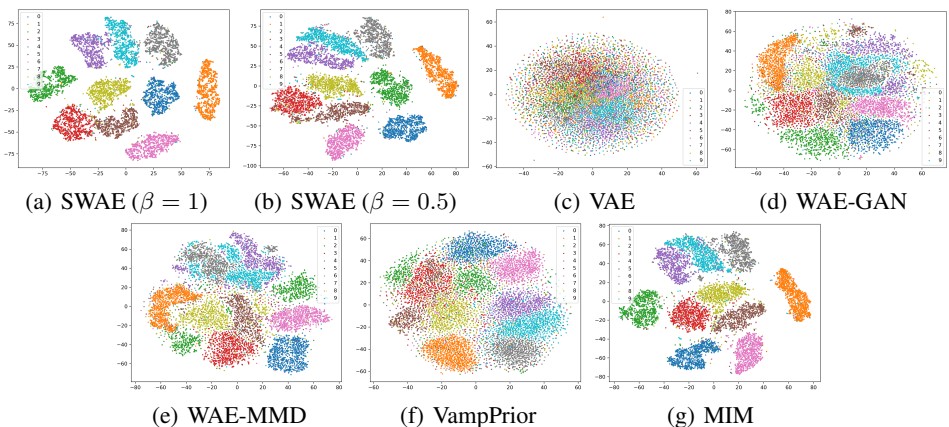

| (a) SWAE ($\beta = 1$) | (b) SWAE ($\beta = 0.5$) | (c) VAE | (d) WAE-GAN |

| (e) WAE-MMD | (f) VampPrior | (g) MIM |

Figure 3: Projection of the latent representation to 2D via t-SNE on MNIST. dim-$\mathbf{z} = 80$ for all methods.

Table 2: Fréchet Inception Distance (FID) on generated images (smaller is better).

| Dataset | dim-$\mathbf{z}$ | SWAE ($\beta = 1$) | SWAE ($\beta = 0.5$) | SWAE ($\beta = 0$) | SWAE ($\beta^*$) | VAE | WAE-GAN | WAE-MMD | VampPrior | MIM |
|---|---|---|---|---|---|---|---|---|---|---|
| MNIST | 8 | 50 | 40 | 24 | 21 | 24 | **17** | 34 | 24 | 74 |
| Fashion-MNIST | 8 | 65 | 57 | 48 | 47 | 60 | **41** | 100 | 51 | 83 |
| Coil20 | 80 | 97 | **89** | 102 | **89** | 278 | 278 | 320 | 97 | 113 |
| CIFAR10-sub | 512 | 105 | **44** | 183 | **44** | 242 | 114 | 341 | 68 | 59 |

## 3.3 GENERATION AND RECONSTRUCTION

To generate new data, latent samples are first drawn from the marginal prior distribution $p(\mathbf{z}_d)$ based on the conditional priors $p(\mathbf{z}_d|\mathbf{u}_k)$, and then fed to the decoder. We put the generated images of all methods in Appendix A.4, and show the Fréchet Inception Distance (FID) (Heusel et al., 2017), which is commonly used for evaluating the quality of generated images, in Table 2. For SWAEs, we observe that the reconstruction loss term is crucial for improving the generation quality as SWAE ($\beta = 1$) generally cannot lead to the lowest FID. On MNIST and Fashion-MNIST, the FID of the best SWAE (indicated as $\beta^*$) is slightly higher than that of WAE-GAN, but lower than all the other benchmarks. The visual difference between SWAE ($\beta^*$) and WAE-GAN on MNIST and Fashion-MNIST is however negligible. In Section 2.1, we compare the formulation of SWAEs ($\beta = 1$) with WAE. In particular, the objective of WAE includes a distribution-based dissimilarity in the latent space while the $\mathbf{z}$-loss in SWAEs measures the sample-based dissimilarity. On Coil20 and CIFAR10-sub, SWAE ($\beta^*$) achieves the lowest FID and generates new images that are visually much better than those generated by the benchmarks.

In Table 3, we compare the reconstruction loss, defined as $\|\mathbf{x}_e - D(\mathbf{z}_e)\|_2^2$, on the four datasets. As expected, increasing the value of dim-$\mathbf{z}$ can reduce the reconstruction loss but the reduction becomes marginal when dim-$\mathbf{z}$ is large enough. Additionally, since a smaller value of $\beta$ leads to more emphasis on the reconstruction-based loss the quality of reconstruction is generally better. We observe that SWAE ($\beta = 0.5$) results in the lowest reconstruction loss in all cases. The reconstructed images of all methods are provided in Appendix A.4 for reference. Without including the reconstruction loss into the objective, the reconstruction quality of SWAE ($\beta = 1$) can be unsatisfactory (*e.g.*, on CIFAR10-sub).

## 3.4 DENOISING EFFECT WITH SWAE ($\beta = 1$)

As discussed in Section 2.1, the $\mathbf{x}$-loss has a close relationship to the objective of Denoising Autoencoders (DAs). After training, we feed the noisy images, which are obtained by adding the Gaussian random samples with mean zero and standard deviation $0.3$ to the clean test samples, to the encoder. In Figure 4, as an example, we show the reconstructed images on Fashion-MNIST. Since the reconstruction loss is highly related to the dimension of the latent space, for fair comparison we set dim-$\mathbf{z}$

Table 3: Reconstruction loss (averaged over 5 trials).

| Dataset | dim-z | SWAE ($\beta = 1$) | SWAE ($\beta = 0.5$) | VAE | WAE-GAN | WAE-MMD | VampPrior | MIM |
|---|---|---|---|---|---|---|---|---|
| MNIST | 8 | $30.11 \pm 0.14$ | $\mathbf{23.20 \pm 0.04}$ | $24.34 \pm 0.07$ | $26.86 \pm 0.37$ | $24.76 \pm 0.31$ | $24.05 \pm 0.10$ | $24.04 \pm 0.10$ |
| | 40 | $26.29 \pm 0.13$ | $\mathbf{6.93 \pm 0.05}$ | $18.40 \pm 0.08$ | $16.06 \pm 0.15$ | $13.78 \pm 0.77$ | $17.32 \pm 0.09$ | $18.14 \pm 0.33$ |
| | 80 | $26.10 \pm 0.09$ | $\mathbf{1.25 \pm 0.02}$ | $18.50 \pm 0.11$ | $10.78 \pm 0.11$ | $9.63 \pm 0.05$ | $17.42 \pm 0.06$ | $17.29 \pm 0.20$ |
| Fashion-MNIST | 8 | $74.74 \pm 0.04$ | $\mathbf{71.03 \pm 0.06}$ | $72.56 \pm 0.02$ | $78.17 \pm 1.41$ | $74.50 \pm 0.60$ | $72.20 \pm 0.04$ | $72.34 \pm 0.03$ |
| | 40 | $73.39 \pm 0.08$ | $\mathbf{57.90 \pm 0.25}$ | $69.85 \pm 0.04$ | $74.84 \pm 0.23$ | $75.86 \pm 0.41$ | $68.67 \pm 0.07$ | $70.22 \pm 0.87$ |
| | 80 | $73.35 \pm 0.08$ | $\mathbf{44.30 \pm 0.71}$ | $69.90 \pm 0.08$ | $70.74 \pm 1.16$ | $71.28 \pm 3.80$ | $68.54 \pm 0.10$ | $69.10 \pm 0.13$ |
| Coil20 | 8 | $7.07 \pm 0.64$ | $\mathbf{5.69 \pm 0.51}$ | $7.90 \pm 0.36$ | $8.14 \pm 0.34$ | $21.20 \pm 15.30$ | $8.17 \pm 1.02$ | $13.84 \pm 3.82$ |
| | 40 | $5.52 \pm 0.40$ | $\mathbf{4.27 \pm 0.66}$ | $5.67 \pm 0.42$ | $5.82 \pm 0.84$ | $8.07 \pm 7.80$ | $6.31 \pm 0.62$ | $5.75 \pm 0.77$ |
| | 80 | $5.56 \pm 0.30$ | $\mathbf{4.33 \pm 0.40}$ | $5.71 \pm 0.67$ | $5.62 \pm 1.26$ | $5.83 \pm 1.92$ | $6.32 \pm 0.37$ | $5.87 \pm 0.69$ |
| CIFAR10-sub | 512 | $50.82 \pm 3.78$ | $\mathbf{6.50 \pm 0.08}$ | $9.41 \pm 0.27$ | $13.37 \pm 1.62$ | $13.09 \pm 1.92$ | $12.06 \pm 0.91$ | $10.02 \pm 0.37$ |

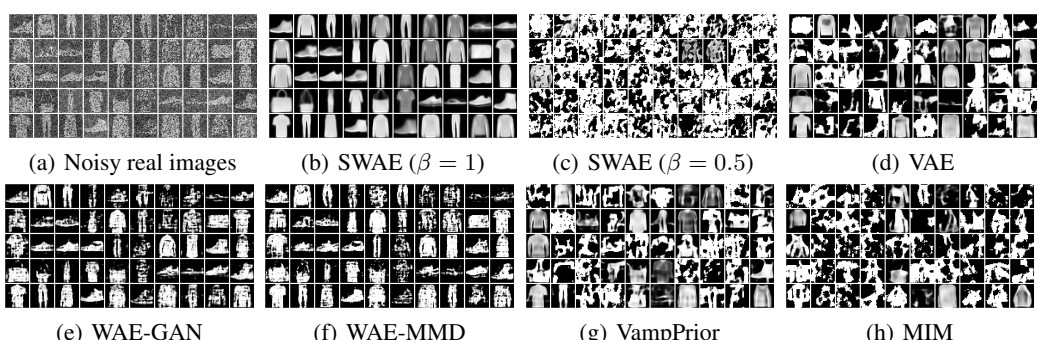

(a) Noisy real images   (b) SWAE ($\beta = 1$)   (c) SWAE ($\beta = 0.5$)   (d) VAE

(e) WAE-GAN   (f) WAE-MMD   (g) VampPrior   (h) MIM

Figure 4: Denoising effect: reconstructed images on Fashion-MNIST. dim-$\mathbf{z} = 80$ for all methods.

to 80 for all methods. We observe that only SWAE ($\beta = 1$) can recover clean images. This observation confirms the denoising effect induced by the $\mathbf{x}$-loss, and thus the resultant latent representation is robust to partial destruction of the observed data.

## 4  RELATED WORK

The objective of VAEs uses the asymmetric KL divergence between the encoding and the decoding distributions (see Appendix A.1). To improve VAEs (Livne et al., 2019; Chen et al., 2018; Pu et al., 2017b) propose symmetric divergence measures instead of the asymmetric KL divergence in VAE-based generative models. For example, MIM (Livne et al., 2019) adopts the Jensen-Shannon (JS) divergence between the encoding and the decoding distributions together with a regularizer maximizing the mutual information between the data and the latent representation. Due to the difficulty of estimating the mutual information and the unavailability of the data distribution, an upper bound of the desired loss is proposed. AS-VAE (Pu et al., 2017b) and the following work (Chen et al., 2018) propose a symmetric form of the KL divergence optimized with adversarial training. These methods typically involve a difficult objective either depending on (unstable) adversarial training or containing the mutual information that requires further approximation. In contrast, the proposed SWAEs yield a simple expression of objective and do not involve adversarial training.

Compared to VAEs, GANs lack an efficient inference model thus are incapable of providing the corresponding latent representation given the observed data. To bridge the gap between VAEs and GANs, recent works attempt to integrate an inference mechanism into GANs by symmetrically treating the observed data and the latent representation, *i.e.*, the discriminator is trained to discriminate the joint samples in both the data and the latent spaces. In particular, the JS divergence between the encoding and the decoding distributions is deployed in ALI (Dumoulin et al., 2017) and BiGANs (Donahue et al., 2017). To address the non-identifiability issue in ALI (*e.g.*, unfaithful reconstruction), later ALICE (Li et al., 2017a) proposes to regularize ALI using conditional entropy.

Generative modelling is closely related to minimizing a dissimilarity measure between two distributions. As opposed to many other commonly adopted dissimilarity measures, *e.g.*, the JS and the KL divergences, the Wasserstein distances that arise from the OT problem provide a weaker distance

between probability distributions (see (Santambrogio, 2015; Peyré et al., 2019; Kolouri et al., 2017) for more background on OT). This is crucial as in many applications the observed data are essentially supported on a low dimensional manifold. In such cases, common dissimilarity measures may fail to provide a useful gradient for training. Consequently, the Wasserstein distances have received a surge of attention for learning generative models (Arjovsky et al., 2017; Balaji et al., 2019; Sanjabi et al., 2018; Kolouri et al., 2019; Patrini et al., 2019; Tolstikhin et al., 2018; Deshpande et al., 2019; Nguyen et al., 2020). Particularly, the VAE-based models (Tolstikhin et al., 2018; Kolouri et al., 2019; Patrini et al., 2019) are all based on minimizing the OT cost of the marginal distributions in the data space with the difference of how to measure the divergence in the latent space: (Tolstikhin et al., 2018) proposes the GAN-based and the MMD-based divergences, (Kolouri et al., 2019) adopts the sliced-Wasserstein distance, and (Patrini et al., 2019) exploits the Sinkhorn divergence. Unlike these works, our proposed SWAEs directly minimize the OT cost of the joint distributions of the observed data and the latent representation with the inclusion of a reconstruction loss for further improving the generative model.

## 5 CONCLUSION AND FUTURE WORK

We contributed a novel family of generative autoencoders, termed Symmetric Wasserstein Autoencoders (SWAEs) under the framework of OT. We proposed to symmetrically match the encoding and the decoding distributions with the inclusion of a reconstruction loss for further improving the generative model. We conducted empirical studies on benchmark tasks to confirm the superior performance of SWAEs over state-of-the-art generative autoencoders.

We believe that symmetrically aligning the encoding and the decoding distributions with a proper regularizer is crucial to improving the performance of generative models. To further enhance the performance of SWAEs, it is worthwhile to exploit other methods for the prior design, *e.g.*, the flow-based approaches (Rezende & Mohamed, 2015; Dinh et al., 2014; 2016), and other forms of the reconstruction loss, *e.g.*, the cross entropy.

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

# A  APPENDIX

## A.1  OBJECTIVE OF VAES

The objective of VAEs is to maximize a tractable variational lower bound on the data log-likelihood, called the Evidence Lower Bound (ELBO):

$$\mathbb{E}_{p(\mathbf{x}_e)}\left[\mathbb{E}_{p(\mathbf{z}_e|\mathbf{x}_e)}[\log p(\mathbf{x}_d|\mathbf{z})] - \mathcal{D}_{\mathrm{KL}}(p(\mathbf{z}_e|\mathbf{x}_e)||p(\mathbf{z}_d))\right]. \tag{6}$$

It can be also shown that the objective of VAEs is equivalent to minimizing the KL divergence (or maximizing the negative KL divergence) between the encoding and the decoding distributions (Livne et al., 2019; Esmaeili et al., 2019; Pu et al., 2017b; Chen et al., 2018):

$$-\mathcal{D}_{\mathrm{KL}}(p(\mathbf{x}_e, \mathbf{z}_e)||p(\mathbf{x}_d, \mathbf{z}_d)) = \mathbb{E}_{p(\mathbf{x}_e, \mathbf{z}_e)}\left[\log \frac{p(\mathbf{x}_d, \mathbf{z}_d)}{p(\mathbf{z}_e|\mathbf{x}_e)}\right] - \mathbb{E}_{p(\mathbf{x}_e)}[\log p(\mathbf{x}_e)]. \tag{7}$$

The right hand side of equation 7 is only different from equation 6 in terms of a constant, which is the entropy of the observed data.

## A.2  PROOF OF THEOREM 1

The proof extends that of Theorem 1 in (Tolstikhin et al., 2018). In particular, (Tolstikhin et al., 2018) aims to minimize the OT cost of the marginal distributions $p(\mathbf{x}_e)$ and $p(\mathbf{x}_d)$, and the proof there is based on the joint probability of three random variables: the observed data, the generated data, and the latent representation. In contrast, we propose to minimize the OT cost of the joint distributions of the observed data and the latent representation induced by the encoder and the decoder. As a result our proof is based on the joint distribution of four random variables $(\mathbf{x}_e, \mathbf{z}_e, \mathbf{x}_d, \mathbf{z}_d) \in \mathcal{X} \times \mathcal{Z} \times \mathcal{X} \times \mathcal{Z}$. We assume that the joint distribution $p(\mathbf{x}_e, \mathbf{z}_e, \mathbf{x}_d, \mathbf{z}_d)$ satisfies the following three conditions:

1. $\mathbf{e} \triangleq (\mathbf{x}_e, \mathbf{z}_e) \sim p(\mathbf{x}_e)p(\mathbf{z}_e|\mathbf{x}_e)$;
2. $\mathbf{d} \triangleq (\mathbf{x}_d, \mathbf{z}_d) \sim p(\mathbf{z}_d)p(\mathbf{x}_d|\mathbf{z}_d)$; and
3. $\mathbf{x}_d \perp\!\!\!\perp \mathbf{x}_e|\mathbf{z}_d$ (conditional independence).

The first two conditions specify the encoder and the decoder respectively, and the last condition indicates that given the latent prior the generated data and the observed data are independent.

Denote the set of the above joint distributions as $\mathcal{P}(\mathbf{x}_e, \mathbf{z}_e, \mathbf{x}_d, \mathbf{z}_d)$. Obviously, we have $\mathcal{P}(\mathbf{x}_e, \mathbf{z}_e, \mathbf{x}_d, \mathbf{z}_d) \subseteq \mathcal{P}(\mathbf{e} \sim p(\mathbf{e}), \mathbf{d} \sim p(\mathbf{d}))$ due to the third condition. If the decoder is deterministic, $p(\mathbf{x}_d|\mathbf{z}_d)$ is a Dirac distribution thus $\mathcal{P}(\mathbf{x}_e, \mathbf{z}_e, \mathbf{x}_d, \mathbf{z}_d) = \mathcal{P}(\mathbf{e} \sim p(\mathbf{e}), \mathbf{d} \sim p(\mathbf{d}))$. With this result, we can rewrite the objective of the underlying OT problem as follows:

$$\begin{aligned}
W_c(p(\mathbf{e}), p(\mathbf{d})) &= \inf_{\Gamma \in \mathcal{P}(\mathbf{x}_e, \mathbf{z}_e, \mathbf{x}_d, \mathbf{z}_d)} \mathbb{E}_{(\mathbf{e}, \mathbf{d}) \sim \Gamma} \quad c(\mathbf{e}, \mathbf{d}) \\
&= \inf_{\Gamma \in \mathcal{P}(\mathbf{x}_e, \mathbf{z}_e, \mathbf{z}_d)} \mathbb{E}_{(\mathbf{x}_e, \mathbf{z}_e, \mathbf{z}_d) \sim \Gamma} \quad c(\mathbf{e}, \mathbf{d}) \tag{8} \\
&= \inf_{p(\mathbf{z}_e|\mathbf{x}_e),\, p(\mathbf{z}_d|\mathbf{x}_e, \mathbf{z}_e)} \mathbb{E}_{p(\mathbf{x}_e)}\mathbb{E}_{p(\mathbf{z}_e|\mathbf{x}_e)}\mathbb{E}_{p(\mathbf{z}_d|\mathbf{x}_e, \mathbf{z}_e)} \quad c(\mathbf{e}, \mathbf{d}) \tag{9} \\
&= \inf_{p(\mathbf{z}_d|\mathbf{x}_e)} \mathbb{E}_{p(\mathbf{x}_e)}\mathbb{E}_{p(\mathbf{z}_d|\mathbf{x}_e)} \quad c(\mathbf{e}, \mathbf{d}), \tag{10}
\end{aligned}$$

where in equation 8 $\mathcal{P}(\mathbf{x}_e, \mathbf{z}_e, \mathbf{z}_d)$ denotes the set of the joint distributions of $(\mathbf{x}_e, \mathbf{z}_e, \mathbf{z}_d)$ induced by $\mathcal{P}(\mathbf{x}_e, \mathbf{z}_e, \mathbf{x}_d, \mathbf{z}_d)$ and it holds due to the deterministic decoder, and equation 10 holds due to the deterministic encoder.

### A.3 DATASETS AND NETWORK ARCHITECTURES

In this section, we briefly describe the datasets, the network architectures, and the hyperparameters that are used in our training algorithm.

- MNIST: The dataset includes $70,000$ binarized images of numeric digits from 0 to 9, each of the size $28 \times 28$. There are $7,000$ images per class. The training set contains $50,000$ images, the validation set contains $10,000$ images for choosing the best model based on the loss function, and the test set contains $10,000$ images.

- Fashion-MNIST: The dataset includes $70,000$ binarized images of fashion products in 10 classes. This dataset has the same image size and the split of the training, validation, and test sets as in MNIST.

- Coil20: The dataset includes gray-scale images of 20 objects, each image of the size $32 \times 32$. The training set contains $1040$ images, the validation set contains $200$ images for choosing the best model based on the loss function, and the test set contains $200$ images.

- CIFAR10-sub: The CIFAR-10 dataset consists of $60,000$ $32 \times 32$ colour images in 10 classes with $6,000$ images per class. There are $40,000$ training, $10,000$ validation, and $10,000$ test images. We randomly select three classes to form the CIFAR10-sub dataset, namely *bird, cat*, and *ship*.

- Celeba: The Celeba dataset is resized to $64 \times 64$ resolution. The training set contains $162,770$ images, the validation set contains $19,867$ images, and the test set contains $19,962$ images.

Network architecture of SWAE: The building block of the network structure of SWAE is based on VampPrior, called GatedConv2d. GatedConv2d contains two convolutional layers with the gating mechanism utilized as an element-wise non-linearity. The parameters in the function GatedConv2d() represent the number of the input channels, the number of the output channels, kernel size, stride, and padding, respectively. The conditional prior network outputs the mean and the log-variance of a Gaussian distribution, based on which the latent prior is sampled.

- The structure of the encoder network: GatedConv2d(1,32,7,1,3)-GatedConv2d(32,32,3,2,1)-GatedConv2d(32,64,5,1,2)-GatedConv2d(64,64,3,2,1)-GatedConv2d(64,6,3,1,1), followed by one fully-connected layer with no activation function.

- The structure of the conditional prior network: The layers of GatedConv2d are the same as those in the encoder network, which are followed by two fully-connected layers. One produces the mean, and the other produces the log-variance with the activation function Hardtanh.

- The structure of the decoder network: Two fully-connected layers with the gating mechanism, followed by GatedConv2d(1,64,3,1,1)-GatedConv2d(64,64,3,1,1)-GatedConv2d(64,64,3,1,1)-GatedConv2d(64,64,3,1,1), followed by a convolutional layer with the activation function Sigmoid.

The algorithm is trained by Adam with the learning rate $= 0.001$, $\beta_1 = 0.9$, and $\beta_2 = 0.999$.

Setup of the number of the pseudo-inputs $K$: As suggested in (Tomczak & Welling, 2018; Livne et al., 2019) we set the value of $K$ in VampPrior and MIM on MNIST and Fashion-MNIST to 500. We found $K = 500$ is also suitable for VampPrior and MIM on Coil20, CIFAR10-sub, and Celeba. Unlike VampPrior and MIM, for SWAE we found that increasing $K$ improves the performance and we set $K$ to 4000 on MNIST, Fashion-MNIST, CIFAR10-sub, and Celeba. Coil20 is a relatively small dataset and we set $K$ to 500 for SWAE, VampPrior, and MIM.

### A.4 MORE EXPERIMENTAL RESULTS

In this section, we show more experimental results based on the comparison with the benchmarks.

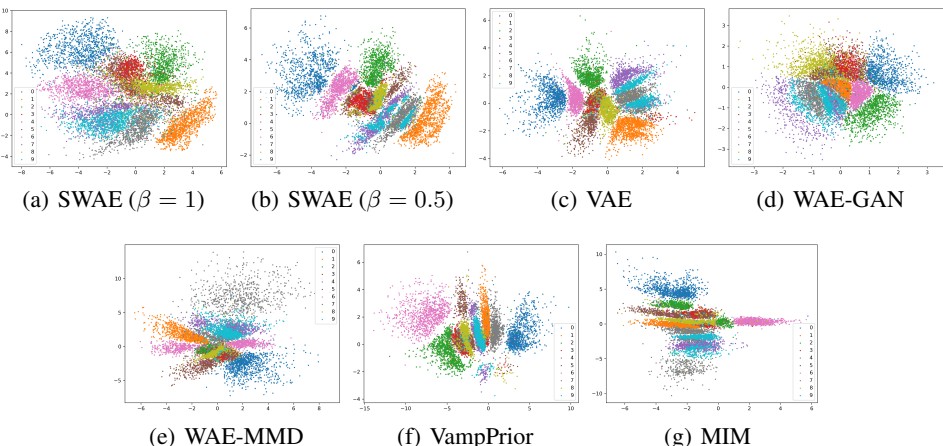

Figure 5: Latent representation on MNIST; dim-$\mathbf{z} = 2$ for all methods. With more compressed latent representation, the classification accuracy generally decreases except VAE (5-NN accuracy with dim-$\mathbf{z} = 2$: SWAE($\beta = 1$)(0.75), SWAE($\beta = 0.5$)(0.86), VAE(0.84), WAE-GAN(0.81), WAE-MMD(0.81), VampPrior(0.82), and MIM(0.87)). Also, the quality of reconstruction and generation decreases when dim-$\mathbf{z} = 2$.

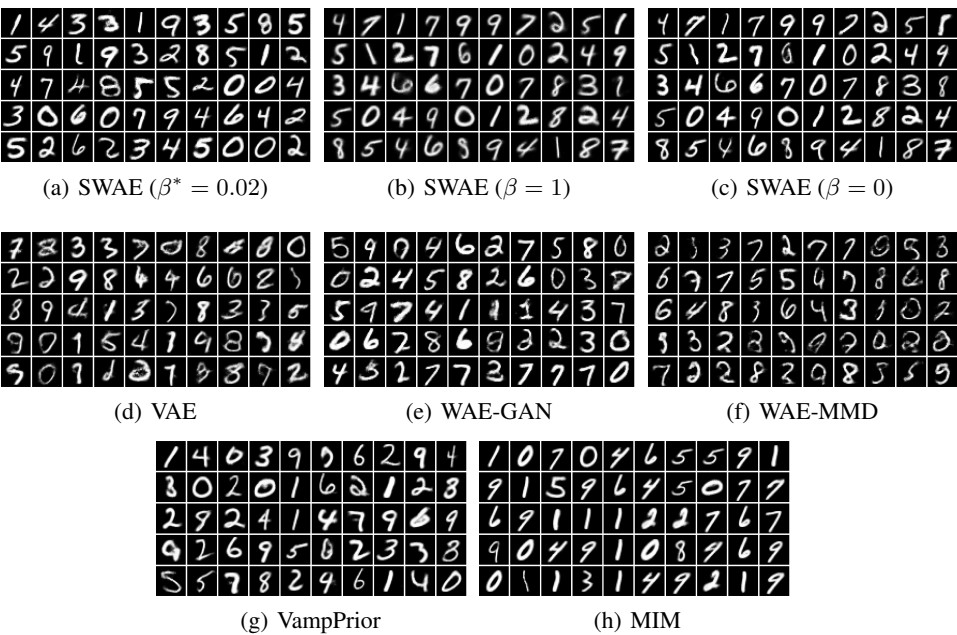

Figure 6: Generated new samples on MNIST. dim-$\mathbf{z} = 8$ for all methods.

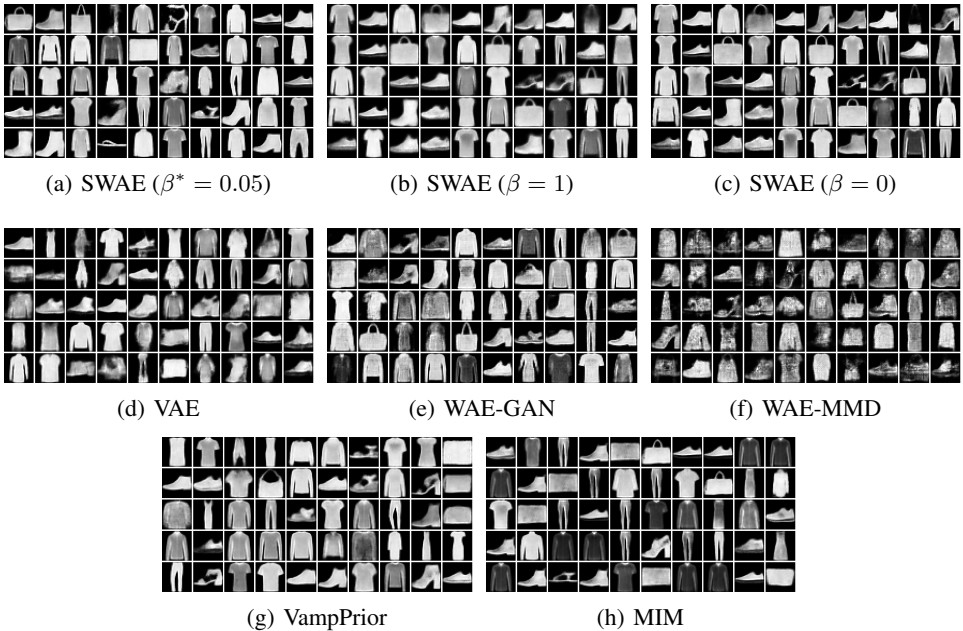

(a) SWAE ($\beta^* = 0.05$)     (b) SWAE ($\beta = 1$)     (c) SWAE ($\beta = 0$)

(d) VAE     (e) WAE-GAN     (f) WAE-MMD

(g) VampPrior     (h) MIM

Figure 7: Generated new samples on Fashion-MNIST. dim-$\mathbf{z} = 8$ for all methods.

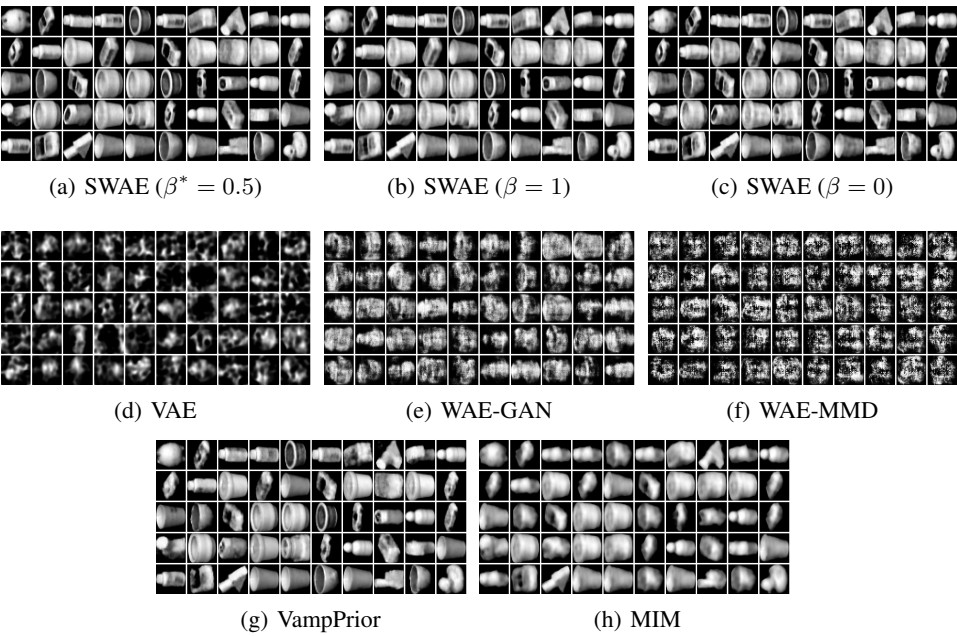

(a) SWAE ($\beta^* = 0.5$)     (b) SWAE ($\beta = 1$)     (c) SWAE ($\beta = 0$)

(d) VAE     (e) WAE-GAN     (f) WAE-MMD

(g) VampPrior     (h) MIM

Figure 8: Generated new samples on Coil20. dim-$\mathbf{z} = 80$ for all methods.

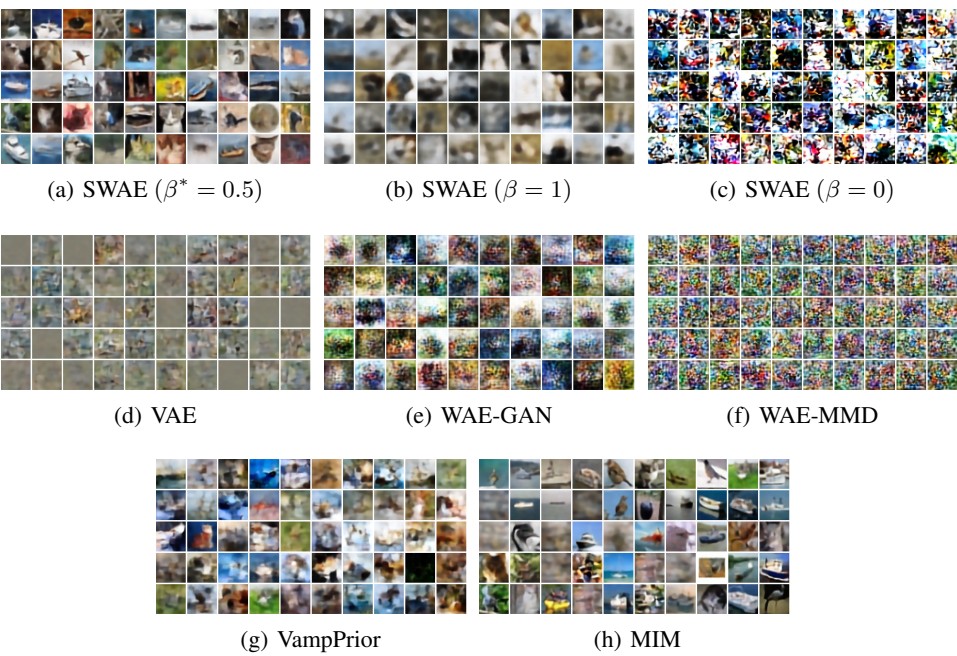

Figure 9: Generated new samples on CIFAR10-sub. dim-$\mathbf{z} = 512$ for all methods.

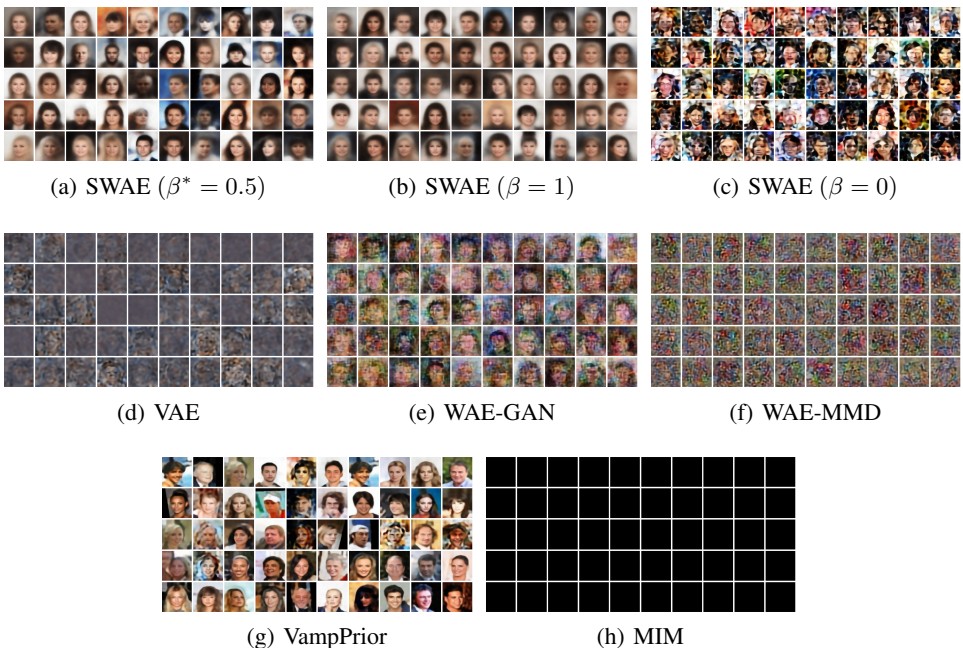

Figure 10: Generated new samples on Celeba. dim-$\mathbf{z} = 512$ for all methods. VampPrior has the best generation quality visually. SWAE ($\beta^* = 0.5$) is better than SWAE ($\beta = 1$) and SWAE ($\beta = 0$).

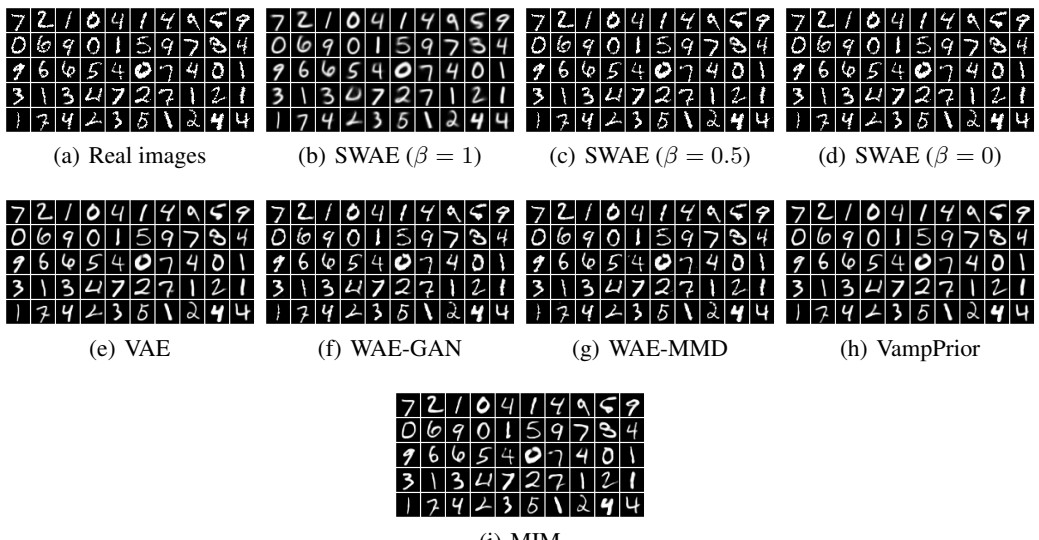

(a) Real images     (b) SWAE ($\beta = 1$)     (c) SWAE ($\beta = 0.5$)     (d) SWAE ($\beta = 0$)

(e) VAE     (f) WAE-GAN     (g) WAE-MMD     (h) VampPrior

(i) MIM

Figure 11: Reconstructed images on MNIST. dim-$\mathbf{z} = 80$ for all methods. As expected, for SWAEs a smaller $\beta$ leads to a higher quality of reconstruction.

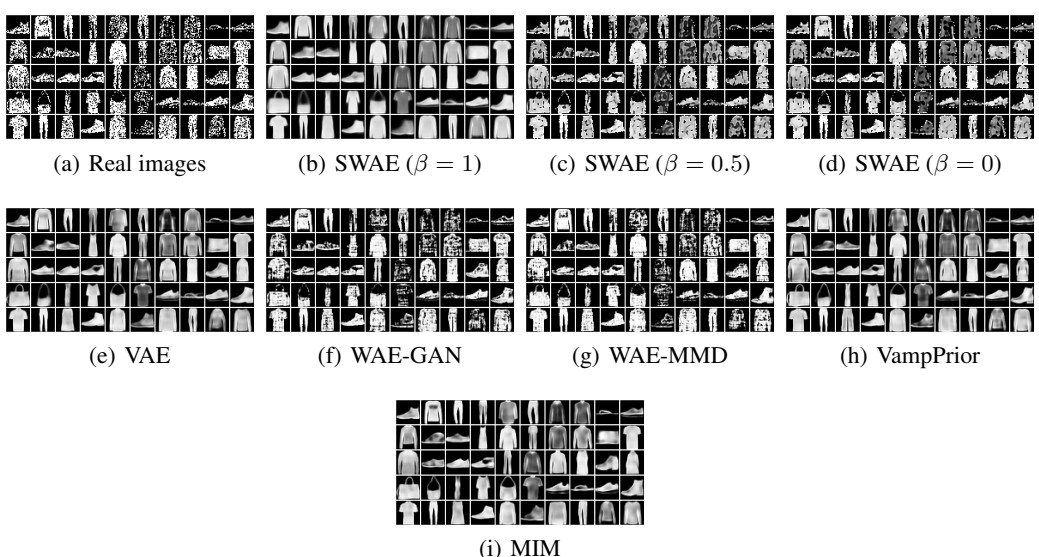

(a) Real images     (b) SWAE ($\beta = 1$)     (c) SWAE ($\beta = 0.5$)     (d) SWAE ($\beta = 0$)

(e) VAE     (f) WAE-GAN     (g) WAE-MMD     (h) VampPrior

(i) MIM

Figure 12: Reconstructed images on Fashion-MNIST. dim-$\mathbf{z} = 80$ for all methods.

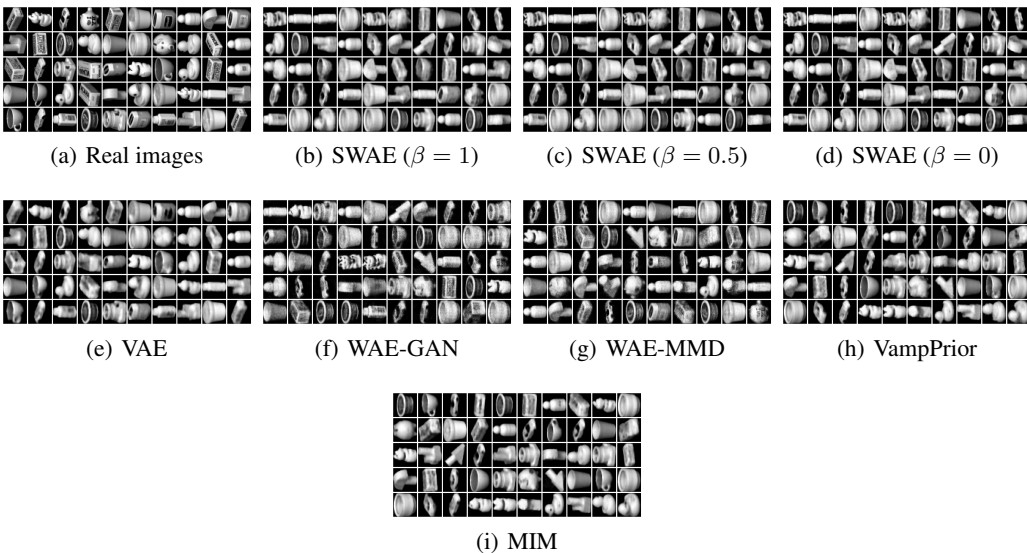

Figure 13: Reconstructed images on Coil20. dim-$\mathbf{z} = 80$ for all methods. For SWAEs, the difference of the reconstruction error for different values of $\beta$ is insignificant, and the reconstructed images look visually the same.

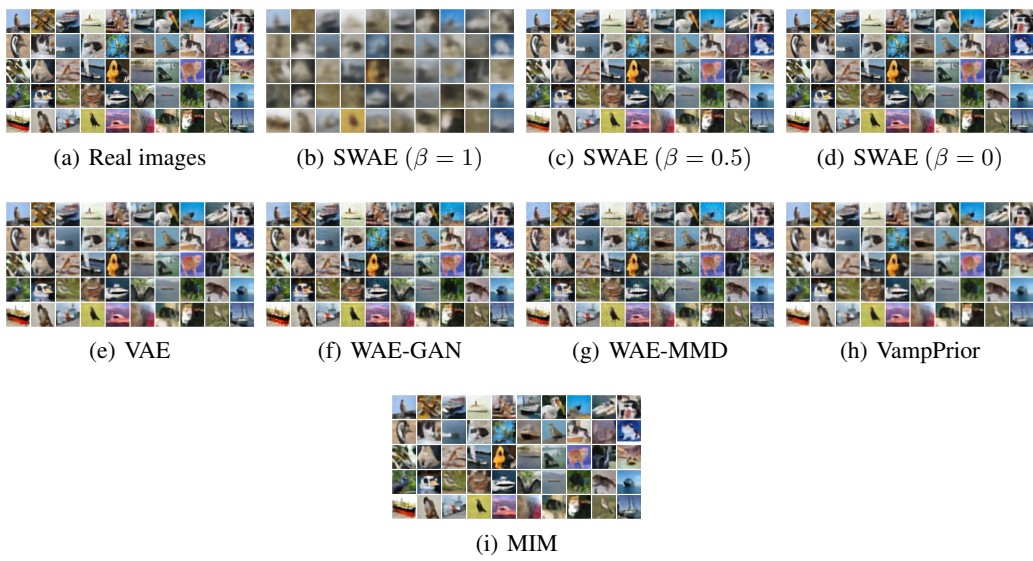

Figure 14: Reconstructed images on CIFAR10-sub. dim-$\mathbf{z} = 512$ for all methods. Excluding the reconstruction loss in the objective, the reconstruction of SWAE ($\beta = 1$) is blurry.

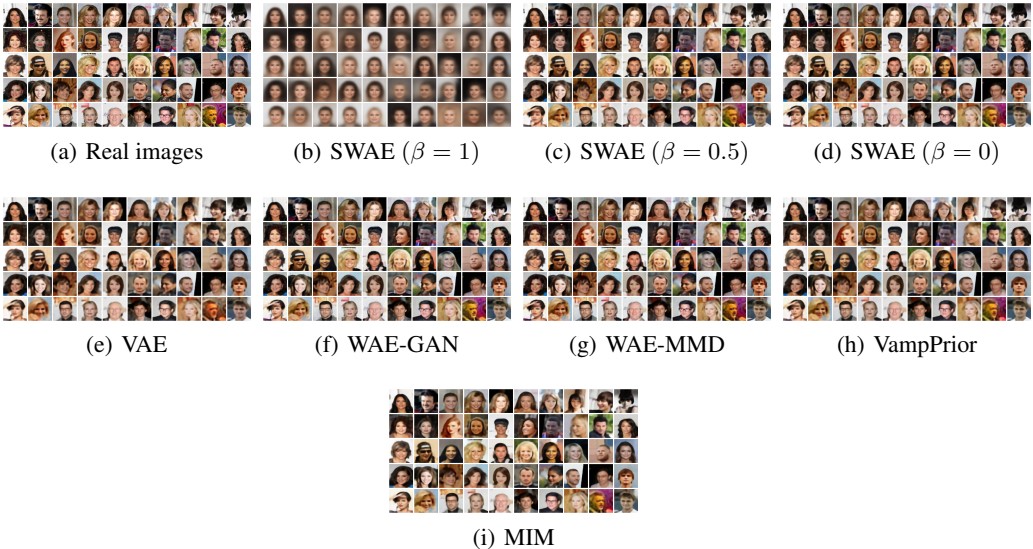

(a) Real images    (b) SWAE ($\beta = 1$)    (c) SWAE ($\beta = 0.5$)    (d) SWAE ($\beta = 0$)

(e) VAE    (f) WAE-GAN    (g) WAE-MMD    (h) VampPrior

(i) MIM

Figure 15: Reconstructed images on Celeba. dim-$\mathbf{z} = 512$ for all methods. The average reconstruction error over three seeds is as follows: SWAE ($\beta = 1$) : 301.11 ±4.85, SWAE ($\beta = 0.5$): 37.78 ±6.71, SWAE ($\beta = 0$): 30.14 ±0.18, VAE: 38.64±5.39, WAE-GAN: **23.63±5.79**, WAE-MMD: 28.64±2.85, VampPrior: 37.96±0.71, and MIM: 36.40±2.84. As expected, for SWAE a smaller value of $\beta$ leads to a lower reconstruction loss.

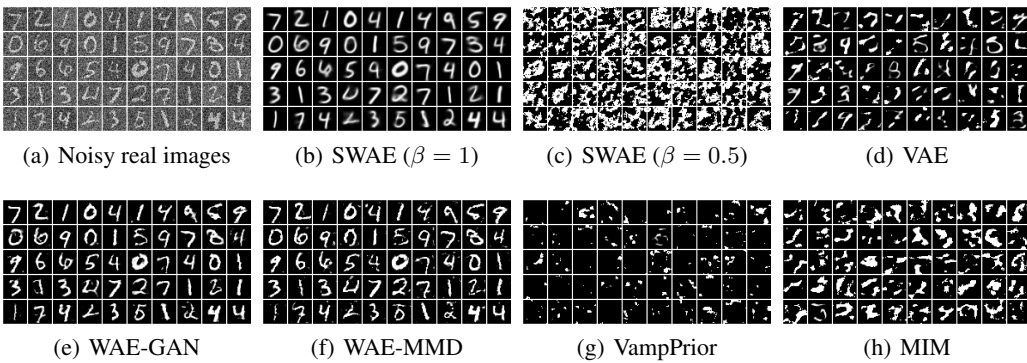

(a) Noisy real images    (b) SWAE ($\beta = 1$)    (c) SWAE ($\beta = 0.5$)    (d) VAE

(e) WAE-GAN    (f) WAE-MMD    (g) VampPrior    (h) MIM

Figure 16: Denoising effect: reconstructed images on MNIST. dim-$\mathbf{z} = 80$ for all methods. SWAE ($\beta = 1$), WAE-GAN, and WAE-MMD can recover clean images. However, for WAE-GAN and WAE-MMD, we can still see some noisy dots around the digits.

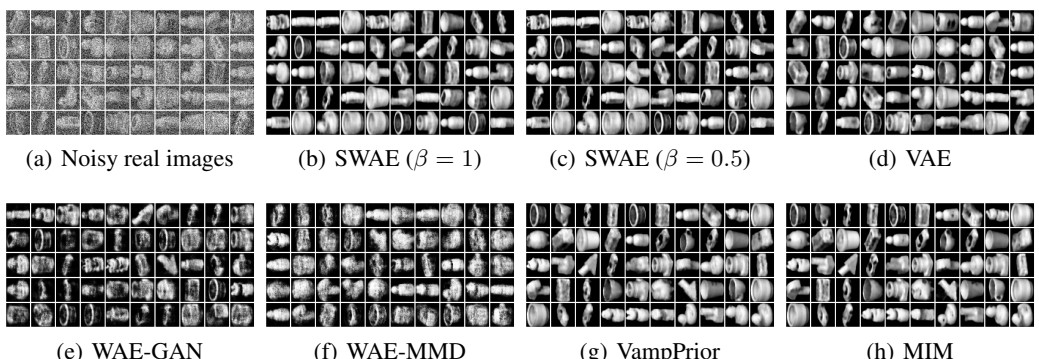

Figure 17: Denoising effect: reconstructed images on Coil20. dim-$\mathbf{z} = 80$ for all methods. Except WAE-GAN and WAE-MMD, the other methods can produce clear images.

