# OpenReview forum: "Symmetric Wasserstein Autoencoders"
_ICLR.cc/2021/Conference — Reject_

### Official Review · AnonReviewer3 · 2020-10-27
**Unclear novelty of Theorem 1 over Tolstikhin 2018 / Empirical evaluations limited to small-scale datasets**

**Rating:** 5
**Confidence:** 4

**Review:**

##########################################################################

Summary:

This works proposes an new auto-encoder variant based on an Optimal Transport (OT) penalty.  While there are many such previous works of OT and auto-encoders, this work proposes a joint OT penalty on data and latent space. As the scalability of computing OT penalties in high dimensions is a concern, the authors address this by restricting to deterministic encoders and decoders in Theorem 1, an extension to joint distributions of Theorem 1 of Tolstikhin 2018. The resulting algorithm amounts to a loss involving L2 penalties for (1) the reconstruction loss (2) decoded latents (conditional on "pseudo-inputs") and real samples (3) encoded samples and the conditional latents. Next experimental results are shown on small-scale datasets (MNIST, Fashion-MNIST, Coil20, subest of CIFAR-10) and compared against the VAE, WAE-{GAN,MMD}, VampPrior, and MIM.

##########################################################################

Reasons for score:

Overall I vote for rejection. Given the numerous prior works (properly cited in the paper), the novelty of the proposed loss appears limited to me. Theorem 1 also appears to be limited novelty over the Theorem 1 of Tolstikhin 2018. Additionally, the empirical evaluation of the method is only limited to small-scale datasets.


##########################################################################

Pros:
* Easy to read
* Reasonable selection of prior models to compare against

##########################################################################

Cons:
* Only small-scale datasets are evaluated. Thus the empirical advantage is unclear
* Theorem 1 appears to be of limited novelty over Theorem 1 in Tolstikin 2018


##########################################################################

Questions during rebuttal period:

Why is your theorem novel over Theorem 1 in Tolstikin 2018?

What is the performance of your method on more complex datasets such as CelebA and LSUN Bedroom?

#########################################################################

POST-REBUTTAL RESPONSE:

Thanks for clarification on Theorem 1 of this paper, i.e. that the novelty is in interpretation. I agree that the "denoising" between observed and generated data is an interesting idea.

I read the author's additional experiments on CelebA. In Figure 10, VampPrior is still qualitatively superior to the author's best result $SWAE(\beta^*=0.5)$. I have some skepticism over the reported results for WAE-{GAN/MMD}, which are much worse than the results in the original paper (Tolstikhin 2018). The authors appear to have used different encoder/decoder architectures, which complicates the comparison. Is WAE's decreased performance due to choice of architecture or algorithm? FID scores on CelebA would be also helpful.

All told, I raise my score, but still harbor some doubts over the empirical advantage of this work.

---

> ### Author Response · Authors · 2020-11-18
> **Explanation on novelty and experiment**
>
> We thank the reviewer for his/her critical comments and we address the questions below.
>
> . The novelty of our proposed SWAEs includes three parts:
>
> 1) fundamentally extend WAEs (Tolstikin 2018) to symmetrically treat the latent and data (Theorem 1). The proof of Theorem 1 is inspired by Theorem 1 in Tolstikin 2018 and is not technically novel. However, the interpretation of Theorem 1 is largely different from that of Theorem 1 in Tolstikin 2018 (see the discussion below Theorem 1 in Section 2.1). For example, the x-loss minimizes the dissimilarity between the observed data and the generated data, and leads to denoising effect. This is different from the reconstruction loss in WAEs and WAEs do not have the denoising effect. Also, due to the symmetric treatment of the latent and data, SWAEs can preserve the local data structure in the latent space.
>
> 2) To improve the latent representation, we additionally incorporated the reconstruction-based loss, which can be balanced with the x-loss via the weighing parameter \beta.
>
> 3) WAEs are optimized over the posterior p(z_e|x_e) with a fixed prior while SWAEs are optimized over the conditional prior p(z_d|x_e) with a learnable prior.
>
> In the original supplementary file we compared SWAEs (\beta = 1) with WAEs in details. In the rebuttal version we moved this discussion to Section 2.1 to emphasize the difference between SWAEs and WAEs.
>
> We will add the experimental results on Celeba soon.

---

> > ### Author Response · Authors · 2020-11-24
> > **Experiment on Celeba**
> >
> > We've added the experimental results on Celeba in the appendix (see Fig. 10 for generation and Fig. 15 for reconstruction). As expected, for generation SWAE(\beta = 0.5) leads to a better quality than both SWAE(\beta = 0) and SWAE(\beta = 1) (as \beta controls the trade-off between x-loss and the reconstruction loss); for reconstruction a smaller value of \beta generally results in a smaller reconstruction loss.
> >
> > For fair comparison in the experiment on Celeba we used similar networks as those for other datasets. Using more advanced networks may further improve the algorithm performance. However, as pointed out in Section 3.1, since the design of neural network architectures is orthogonal to that of the algorithm objective and the latter is our main purpose, we did not exploit other network structures.

---

### Official Review · AnonReviewer1 · 2020-10-28
**The proposal of Symmetric Wasserstein Autoencoders is quite interesting but lacks novelty.**

**Rating:** 5
**Confidence:** 4

**Review:**

In the paper, the authors propose a new family of generative auto-encoders, named Symmetric Wasserstein Autoencoders (SWAEs), based on replacing the KL divergence between the encoding and decoding distributions on the traditional VAE framework into the Wasserstein metric between these distributions. Then, they also carried out experiments to demonstrate the favorable performance of their auto-encoder over previous base-line autoencoders.

I think the SWAEs is quite interesting but lacks novelty. Here are my comments with the paper:

(1) The result of Theorem 1 is under the assumption that both the encoder and decoder are deterministic, which is quite restrictive. Can the authors provide some understandings when either one of them is random? The result of Theorem 1 is also quite similar to the main result in the Wasserstein autoencoder work. I think the authors should at least cite this point properly.

(2) Simply replacing KL divergence between the encoding and decoding distributions by Wasserstein metric to have symmetric properties sounds not novel. The KL divergence has a good interpretation in terms of ELBO. What can we interpret the Wasserstein metric between the encoding and decoding distributions? Theorem 1 does not seem convincing to me.

(3) In the experiments, the choice of $\beta = 1/2$ seems to yield best results, i.e., we should balance the reconstruction loss and the discrepancy in the data space in the objective function (4). Can the authors provide some intuition behind that?

(4) The related work with generative modeling based on optimal transport lacks several relevant recent works; see for examples [1], which achieves SOTA among sliced-based Wasserstein distances in deep generative models, or [2], which proposes max-sliced Wasserstein distance for deep generative models.

References:

[1] K. Nguyen, N. Ho, T. Pham, H. Bui. Distributional sliced-Wasserstein and applications to deep generative modeling. Arxiv preprint Arxiv: 2002.07367, 2020.

[2] I. Deshpande, Y. Hu, R. Sun, A. Pyrros, N. Siddiqui, S. Koyejo, Z. Zhao, D. Forsyth, A. Schwing. Max-sliced Wasserstein distance and its use for GANs. CVPR, 2019.

---

> ### Author Response · Authors · 2020-11-18
> **Explanation on novelty and experiment**
>
> We thank the reviewer for his/her critical comments and we address the questions below.
>
> 1)
> a) The proof of Theorem 1 extends that of Theorem 1 in WAEs paper, which has been explicitly stated in the appendix A.2 Proof of Theorem 1. To make this clear, we've also added this statement in Section 2.1  in the rebuttal reversion.
>
> Although the proof of Theorem 1 is similar to that in WAEs the interpretation of our theorem is largely different from theirs (see the discussion below Theorem 1 in Section 2.1). For example, the x-loss minimizes the dissimilarity between the observed data and the generated data, and leads to denoising effect. This is different from the reconstruction loss in WAEs and WAEs do not have the denoising effect. Also, due to the symmetric treatment of the latent and data SWAEs can preserve the local data structure in the latent space.
>
> In the original supplementary file we compared SWAEs (\beta = 1) with WAEs in details. In the rebuttal version we moved this discussion to Section 2.1 to emphasize the difference between SWAEs and WAEs.
>
> b) The same as WAEs, under the assumption of deterministic encoder/decoder, we can get nice and clean result as in Theorem 1. With random decoders, only an upper bound of the OT cost can be obtained in WAEs (see Appendix B.1 in WAEs). Since we consider the matching of the joint distributions instead of the marginal distributions we expect more complicated theoretical conclusion, which can hardly guide the design of our generative model. In particular, with random encoders the current conditional prior p(z_d|x_e) also depends on the latent representation z_e, which may complicate the design of the prior z_d.
>
> We noticed that some follow-up work tried to relax the deterministic condition in WAEs. For example (Bahuleyan et al 2018) relaxed the deterministic encoder to a stochastic one, which however may lead to collapse to a Dirac delta function and thus extra regularization is required. Therefore the relaxation of SWAEs to a random version is not straightforward and is worthy of more study.
>
> Although with deterministic encoder/decoder, unlike in WAEs where the prior is fixed, in SWAEs the prior is learnable through the conditional prior, which increases the flexibility of the network.
>
> Reference:
> Bahuleyan, H., Mou, L., Zhou, H., & Vechtomova, O. (2018). Stochastic wasserstein autoencoder for probabilistic sentence generation. arXiv preprint arXiv:1806.08462.
>
> 2) Although Wasserstein metric has been widely used in generative models (e.g., VAEs/GANs), as far as we know it has not been applied to model the joint distributions of the latent and data in VAE-based models. Our main contribution is to fundamentally extend WAEs to symmetrically treat the latent and data.
>
> The KL divergence of the joint distributions is equivalent to ELBO in VAEs. However, the objective of VAEs has limitations; in particular the asymmetry of KL often leads to unrealistic generated samples (Li et al 2017b; Alemi et al 2017), which motivates a symmetric divergence metric such as Wasserstein metric. The interpretation of Theorem 1 has been extensively discussed below Theorem 1 in terms of x-loss and z-loss. Specifically, due to the symmetric treatment, the objective in equation 3 preserves the local data structure in the latent space. Roughly speaking, if two data samples are close to each other in the data space, their corresponding latent representations are also expected to be close.
>
> 3) The weighting parameter \beta controls the trade-off between the x-loss and the reconstruction loss. \beta needs to be carefully chosen to achieve a better trade-off between the generation and reconstruction. A smaller value of \beta generally leads to better reconstruction since more emphasis is put on the reconstruction loss (see Figs. 10-13 for reconstructed images). In contrast, for generation the best value of \beta (i.e., \beta^*) may depend on datasets. For example, \beta^* on Coil20 and CIFAR10-sub is 0.5 while \beta^* on MNIST and Fashion-MNIST is not.
> To illustrate the above point we've included the above discussion in Section 2.2 in the rebuttal reversion.
>
> 4) Thanks for bringing these references to our attention. We've included these papers in related work in the rebuttal reversion.

---

### Official Review · AnonReviewer2 · 2020-10-28
**Interesting method, but the experiments need to be refined**

**Rating:** 5
**Confidence:** 4

**Review:**

This paper proposes to treat the encoding and the decoding pairs symmetrically as a solution to OT problems. SWAE minimizes $p(x_d, z_d)$ and $p(x_e, z_e)$ in a jointly manner and shows better latent representation learning and generation. Moreover, the symmetric treatment for encoding and decoding shows an advantage in data denoising.


----------

It is interesting to minimize the distance between $p(x_d, z_d)$ and $p(x_e, z_e)$ with the OT formulation.  The usage of the deterministic encoder and decoder could solve the problem of $W(p_{x_e}, p_{e_d})$ minimization, while it is difficult for the latent code since the latent codes from the prior and the posterior are not paired. Here it is good to see the authors apply aggregated posterior methods to solve this. The usage of the closest pseudo-inputs in constructing the pair of $z_e$ and $z_d$ could be a good approximation for $W(p_{z_e}, p_{z_d})$.

My questions mainly lie in the experiment part:

The authors report SWAE with three configurations, where $\beta=1$, $\beta=0.5$ and $\beta=0$ respectively. When $\beta \rightarrow 1$, SWAE is only minimizing the $d( p(x_d, z_d), p(x_e, z_e) ) $; when $\beta \rightarrow 0$, SWAE behaves like an autoencoder with regularization in latent space.  The classification results and the reconstruction results of the case $\beta = 0$  are missing. I am curious how these two cases perform, are they perform like VAE and Vampprior VAE?

The authors visualize the latent space with t-sne where the dimensionality of $z$ is 80 before the dimension reduction. As far as I know, when the dimension of $z$ is large enough, the t-sne results of VAEs and WAEs are similar to figure 3a, 3b and 3g. The results shown in c,d,e and f seem degenerated.  Especially 3c, it does not look like what we have seen in previous papers. For example, figure 2 in Makhzani et al, 2015 (https://arxiv.org/pdf/1511.05644.pdf) shows that the latent representations produced by VAE have several modes.  I doubt this could be the problem of dimension reduction. It will be more convincing if the author could show the visualization of these models where $dim(z) = 2$ and without any dimension reduction.

For the reconstruction results (not the denoising reconstruction) shown in Figure 9 and 10, the images seem to be dynamically binarized. However, in the description of the experiment details, this part is not mentioned. Considering SWAE is not applying a Bernoulli decoder, the dynamic binarization is not necessary. Moreover, the reconstruction from SWAE ($\beta=0.5$) and SWAE ($\beta=0$) seem to be binarized, which is not reasonable.

When $\beta=1$, a good baseline for SWAE will be ALI (Dumoulin et al., 2017), since ALI is minimizing $JS(p(x_d, z_d)|| p(x_e, z_e))$. I am curious how SWAE ($\beta=1$) outperforms/underperforms compared to ALI and some corresponding analysis from the authors will be appreciated.

In the random generation results, only SWAE ($\beta^\star$) is shown. The generation results by SWAE ($\beta=1$) and SWAE ($\beta=0$) are missing while the quantitative results are shown in table 2.


---------------

In general, the idea of symmetrically treating encoding and decoding to solve with the Wasserstein distance is interesting and is worthy of study. However, some details in the experiments remain unclear; some experiments results and the corresponding analysis are missing. The authors might need to dig out more to support their method.

---

> ### Author Response · Authors · 2020-11-18
> **Explanation on experiment**
>
> We thank the reviewer for his/her critical comments and we address the questions below.
>
> . The classification results of SWAE (\beta = 0) are similar to SWAE(\beta = 1) and  SWAE(\beta = 0.5) on MNIST, Fashion-MNIST, and Coil20, but worse on CIFAR10-sub. We attribute the degraded classification performance of SWAE (\beta = 0) to the lack of symmetric treatment of the latent and data when compared to SWAE(\beta = 1) and  SWAE(\beta = 0.5).
> We've added the classification results of SWAE (\beta = 0) in the rebuttal version (Table 1).
>
> . The reconstruction loss of SWAE (\beta = 0) is even lower than SWAE (\beta = 0.5) since SWAE (\beta = 0) emphasizes more on the reconstruction loss. Hence the reconstruction quality of SWAE (\beta = 0) is better than SWAE (\beta = 0.5) and also VAE and VampPrior. However, the generation quality of SWAE (\beta = 0) may be worse. SWAE (\beta = 0) does not perform like VAE or VampPrior as their objective functions are different.
> We've added this discussion in Section 3.3 in the rebuttal reversion.
>
> . latent visualization with dim-z = 2: We've added the latent visualization with dim-z = 2 in the appendix of the rebuttal version (see Fig. 5). With more compressed latent representation, the classification accuracy generally decreases except VAE (5-NN accuracy with dim-z = 2: SWAE(\beta = 1)(0.75), SWAE(\beta = 0.5)(0.86), VAE(0.84), WAE-GAN(0.81), WAE-MMD(0.81), VampPrior(0.82), and MIM(0.87)). Also, the quality of reconstruction and generation decreases when dim-z = 2.
>
> . dynamic-MNIST and dynamic-Fashion MNIST: Following the benchmarks VampPrior and MIM, we binarized MNIST and Fashion MNIST for fair comparison. Since we focus on deterministic encoder/decoder, instead of a Bernoulli decoder we equipped the last layer of the decoder with the Sigmoid activation function. Following VampPrior, the pseudo-inputs are initialized with random training samples before the binarization. We will revise the description of the dataset in the appendix  accordingly.
> Since the training samples in Figs. 9 and 10 are binarized, to achieve better reconstruction quality, the reconstruction from SWAE(\beta = 0.5) and SWAE(\beta = 0) also tends to be binarized. So this observation is reasonable.
>
> . Comparison with ALI(Dumoulin et al., 2017): We've discussed ALI in Section 4 related work. ALI is related to our work as it also symmetrically treats the latent and data, and it uses the symmetric JS divergence. However, ALI is a GAN-based model while our proposed SWAE is VAE-based. Therefore, the comparison may not be straightforward. In experiment, we compared to MIM, which is VAE-based and also employs JS divergence between the encoding and decoding distributions. For fair comparison, the network architecture of MIM and SWAE is  built similarly. The experimental results show that SWAE is superior to MIM in general.
>
> . Generation of SWAE (\beta = 1) and SWAE (\beta = 0): We've added the generated samples of SWAE (\beta = 1) and SWAE (\beta = 0) in the appendix for comparison.

---

### Official Review · AnonReviewer4 · 2020-10-31
**A nice generative autoencoder using optimal transport**

**Rating:** 6
**Confidence:** 4

**Review:**

This paper proposed symmetric Wasserstein autoencoders (SWAE), which is a new type of generative autoencoders within the framework of optimal transport (OT). This work is based on Wasserstein autoencoders (WAE), but leverage a symmetric distance between the encoding distribution and the decoding distribution to better preserve the local structure of the data in the latent space. A reconstruction loss based regularization is suggested for better performance. The paper is well written and easy to follow. My only concern is that the improvement over WAE seems to be marginal. Below are some minor issues that need to be addressed.

1. Please provide the appendix.

2. The data sets used in the experiment are kind of too simple. Can the author provide results for full CIFAR10?

3. The FID of WAE-GAN and WAE-MMD in Table 2 does not look right, can the author provide implementation info? Did you use your own code?

---

> ### Author Response · Authors · 2020-11-18
> **Improvement over WAEs and other minor issues**
>
> We thank the reviewer for his/her critical comments and we address the questions below.
>
> Our proposed SWAEs is fundamentally different from WAEs:
>
> 1) SWAEs aim at minimizing the OT cost between the joint distributions of the latent and data while WAEs minimize the OT cost between the data distributions. Although the proof of Theorem 1 is similar to that in WAEs
> the interpretation of our theorem is largely different from theirs (see the discussion below Theorem 1 in Section 2.1). For example, the x-loss minimizes the dissimilarity between the observed data and the generated data, and leads to denoising effect. This is different from the reconstruction loss in WAEs and WAEs do not have the denoising effect. Also, due to the symmetric treatment of the latent and data SWAEs can preserve the local data structure in the latent space.
>
> 2) To improve the latent representation we additionally incorporated the reconstruction-based loss, which can be balanced with the x-loss via the weighing parameter \beta.
>
> 3) WAEs are optimized over the posterior p(z_e|x_e) with a fixed prior while SWAEs are optimized over the conditional prior p(z_d|x_e) with a learnable prior.
>
> The experimental results also demonstrated the superiority of SWAEs to WAEs in terms of classification accuracy, latent representation, generation, and reconstruction. In the original supplementary file A.3 we compared SWAEs (\beta = 1) with WAEs in details. In the rebuttal version we moved this discussion to Section 2.1 to emphasize the difference between SWAEs and WAEs.
>
> 1. The appendix was originally included in the supplementary file for the initial submission. To facilitate reading, we now combine the main file with the appendix.
>
> 2. CIFAR10-dataset:  For fair comparison, in experiment we built SWAE based on the VampPrior network architecture. As explained in Section 3.2, we found that the classification results of all algorithms on CIFAR10 are unsatisfactory based on the current networks (accuracy was around 0.3−0.4; this may due to the limited expressive power of the shallow network architectures used). Using more advanced network may improve the model performance. However, as pointed out in the beginning of Section 3.1, the  design  of  neural  network  architectures  is  orthogonal to that of the algorithm objective (Vahdat & Kautz, 2020), hence is out of the scope of this paper.
>
> 3. We did not use our own code for WAEs. We used the code released at https://github.com/schelotto/Wasserstein-AutoEncoders.

---

### Author Response · Authors · 2020-11-24
**Discussion phase ends soon.**

Dear Reviewers,

We appreciate your insightful comments on our paper.

Since the second discussion phase will end soon, it would be much appreciated if you could let us know if you have any further comments or concerns that we have not addressed up to your satisfactory.  We will be happy to address them to improve our paper.

Thank you!

---

### Decision · Program_Chairs · 2021-01-07
**Final Decision**

**Decision:**

Reject

**Comment:**

In this paper, the authors proposed a new variant of the Wasserstein autoencoder (WAE), which matches the joint distribution of data and the latent codes induced by the encoder and the joint distribution induced by the decoder in the framework of optimal transport. Because of matching the distributions that are not considered by existing autoencoders like WAEs or VAEs, I agree with the authors that the proposed method is novel to some degrees.

However, the experimental part does not support the superiority of the method well. For example, some reviewers (including me) think the results of the baselines shown in Figures 8-10 are underestimated. According to my personal experience, the WAE should perform much better on CelebA than that shown in Figure 10. The experiments in Figures 11-17 provide more reasonable results, but the advantage of the proposed method is not convincing.

Here is my suggestion:
1) Because the proposed method can achieve flexible prior, besides randomly generating data, the authors can consider adding some experiments on conditional generation, i.e., generating data from a single modality of the learned prior. I believe the proposed method will be more convincing if it can show some advantages in the conditional generation task.
2) The runtime comparison for the method and the baselines in the training phase should be discussed.
3) The short name "SWAE" is in conflict with an earlier work "Sliced Wasserstein Autoencoder", which is also called "SWAE".